# Lightweight Model for Pavement Defect Detection Based on Improved YOLOv7

**DOI:** 10.3390/s23167112

**Published:** 2023-08-11

**Authors:** Peile Huang, Shenghuai Wang, Jianyu Chen, Weijie Li, Xing Peng

**Affiliations:** School of Mechanical Engineering, Hubei University of Automotive Technology, Shiyan 442002, China; 202210001@huat.edu.cn (P.H.); 202010004@huat.edu.cn (J.C.); 202211089@huat.edu.cn (W.L.); 202210010@huat.edu.cn (X.P.)

**Keywords:** pavement defect detection, defect detection, YOLOv7, SPPCSPC_Group, K-means, Ghost Conv module, CBAM convolution module

## Abstract

Existing pavement defect detection models face challenges in balancing detection accuracy and speed while being constrained by large parameter sizes, hindering deployment on edge terminal devices with limited computing resources. To address these issues, this paper proposes a lightweight pavement defect detection model based on an improved YOLOv7 architecture. The model introduces four key enhancements: first, the incorporation of the SPPCSPC_Group grouped space pyramid pooling module to reduce the parameter load and computational complexity; second, the utilization of the K-means clustering algorithm for generating anchors, accelerating model convergence; third, the integration of the Ghost Conv module, enhancing feature extraction while minimizing the parameters and calculations; fourth, introduction of the CBAM convolution module to enrich the semantic information in the last layer of the backbone network. The experimental results demonstrate that the improved model achieved an average accuracy of 91%, and the accuracy in detecting broken plates and repaired models increased by 9% and 8%, respectively, compared to the original model. Moreover, the improved model exhibited reductions of 14.4% and 29.3% in the calculations and parameters, respectively, and a 29.1% decrease in the model size, resulting in an impressive 80 FPS (frames per second). The enhanced YOLOv7 successfully balances parameter reduction and computation while maintaining high accuracy, making it a more suitable choice for pavement defect detection compared with other algorithms.

## 1. Introduction

With the continuous development of society, the investment in highways in our country has been on the rise. As of the end of 2021, the total mileage of the road network has reached an impressive 5.2807 million km, with expressways covering 169,100 km [1], making it one of the world’s leading road networks. This extensive road network greatly facilitates information exchange and resource allocation among regions, contributing significantly to rapid economic and social development. However, the surge in vehicles utilizing these roads has led to an increase in road usage frequency, resulting in various defects on the road surface. These defects can negatively impact the lifespan of the road and may even lead to traffic accidents in severe cases. To address these issues, management authorities require regular and scientific testing of road surfaces to promptly detect and address any defects.

Traditional artificial vision detection of pavement issues is a complex and labor-intensive process with low detection efficiency. Nonetheless, advancements in photography technology and deep learning have offered technical support for image collection and recognition of road surface defects. This paper builds upon the YOLOv7 target detection algorithm [2], a notable achievement in the YOLO series, and aims to make further improvements. While YOLOv7 already exhibits high detection accuracy and inference speed, the model’s parameter count and computational load have increased, hindering practical applications. To overcome these challenges and maintain the detection accuracy while reducing the computations and parameters, this paper proposes a lightweight pavement defect detection model based on an enhanced YOLOv7 architecture. The research builds upon prior work in the field of road surface defect detection.

Ye G. et al. [3] proposed an improved YOLOv7 network, designing and enhancing multiple different self-developed modules to better identify many misleading targets of concrete cracks. Chen J. et al. [4] introduced small object detection layer, lightweight convolution and CBAM (Convolutional Block Attention Module) attention mechanism to achieve multiscale feature extraction and fusion, reducing the number of model parameters. Zhang M. et al. [5] proposed a lightweight underwater target detection method based on MobileNet v2, the You Only Look Once (YOLO) v4 algorithm and attention feature fusion, which provides accuracy and speed for target detection in marine environments. Li C. et al. [6] proposed a lightweight YOLOv7 algorithm model by replacing the backbone network with Mobilenetv89, enlarging the feature map generated by the feature pyramid, and adding a detection head based on the attention mechanism to improve the detection performance of occluded pedestrians and small pedestrians. Long et al. [7] proposed a lightweight convolutional neural network, LiraNet, which is a lightweight model for ship detection in radar images, and it can be easily deployed on mobile devices. Du F. J. et al. [8] used the BIFPN (Bidirectional Feature Pyramid Network) structure for multiscale feature fusion, with enhanced feature extraction capabilities, and used zoom loss to optimize the sample imbalance problem and improve the detection accuracy of road defect targets. Jiang J. et al. [9] proposed a multichannel fusion SAR image processing method that fully utilizes image information and network feature extraction capabilities; it is also based on the latest You Only Look Once Version 4 (YOLO-V4) deep learning framework to model the architecture and train the model. Wan, F. et al. [10] proposed a new backbone network, Shuffle-ECANet, by enhancing the YOLOv5s method, adding an ECA attention module to the lightweight model, ShuffleNetV2. Fang et al. [11] proposed Tinier-YOLO, derived from Tiny-YOLO-V3, to further reduce the model size while improving the detection accuracy and real-time performance. With the TRC-YOLO, Wang G. et al. [12] pruned the convolution kernel of YOLO v4-tiny and introduced an extended convolution layer in the residual module of the network to generate an hourglass cross-stage part ResNet (CSPResNet) structure. Xia Y. et al. [13] added the attention module to YOLOv7 and increased the weight of the visual features while suppressing the weight of invalid features. Liu Haohan et al. [14] proposed an improved YOLOv7-tiny model, introducing the ShuffleNet v1 network, GSConv (ghost shuffled convolution) module to improve the neck layer of the network, and used the Mish activation function to increase the nonlinear expression and improve the generalization ability of the model. Duan Bichong et al. [15] proposed a mask detection algorithm based on an improved YOLOv5, using the lightweight network GhostNetV2 to replace the C3 module in the YOLOv5s backbone network, introducing the CBAM attention mechanism and the loss function using EIoU to replace GIoU to improve positioning accuracy. Tu Chengfeng et al. [16] introduced the lightweight network ShuffleNetv2 and GhostNet on the YOLOv5n architecture to realize the lightweight detection network. D. Ma et al. [17] proposed a tracking network composed of pavement crack generation confrontation network (PCGAN) and crack detection YOLO-MF. The system can detect and track cracks in real time when deployed on equipment. Kaya Ö et al. [18] proposed to use the faster R-CNN (R101-FPN and X101-FPN) and YOLOv7 network model to design the automatic detection of pedestrian crosswalks in urban road networks from the perspective of pedestrians and vehicles. Que Y. et al. [19] proposed a generative adversarial network (GAN)-based method for the data augmentation of collected digital images of cracks and proposed an improved deep learning network (i.e., VGG) for crack classification.

Previous research in the domain of lightweight models mainly involved using lightweight networks or directly employing pruning techniques. However, these approaches often faced accuracy loss after lightweighting. To ensure detection accuracy while reducing the computation and parameters, this paper proposes a multifaceted approach for pavement defect detection based on an improved YOLOv7 model. The enhancements include introducing the SPPCSPC_Group (grouped space pyramid pooling module) to reduce the model parameters, employing the K-means clustering algorithm for anchor regeneration to expedite model convergence, incorporating the Ghost Conv module as a new feature extraction network to improve detection effectiveness while minimizing the parameters and calculations and, finally, introducing the CBAM convolution module to enhance the network’s global information learning.

## 2. Related Work

### 2.1. YOLOv7

As a typical single-stage target detection algorithm, the YOLO [20,21,22] series is widely used in the real-time detection of the system because of its fast running speed, and YOLOv7 [23] has excellent detection accuracy and detection speed in the YOLO series model, so this paper chose YOLOv7 as the asphalt pavement algorithmic model for defect detection.

The YOLOv7 algorithm can increase the depth of the network by introducing Extended-ELAN (Extended-Efficient Layer Aggregation Network) and strategies such as model scaling and convolution re-parameterization based on concatenation-based models. To improve the accuracy of the network, its model consists of an input terminal, a backbone architecture, and a head architecture, as shown in Figure 1.

First, the input end is the input layer, whose main function is to scale the input image to the same size and then input it to the backbone architecture to meet the training requirements of the backbone network. The backbone architecture is called the feature extraction layer, consisting of 50 layers (Layer 0~50). Different convolution combination modules are composed, and the main function is to extract 3 different sizes of target information features and input them to the head architecture, and the output positions of the backbone architecture are located at the 24th layer, the 37th layer, and the 50th layer, respectively. The head architecture mainly includes the SPPCSPC layer (Spatial Pyramid Pooling Connected Spatial Pyramid Convolution), E-ELAN layer, several Conv layers (Convolution), MP layer (MaxPool), and REPConv layer; its biggest feature is the use of efficient E-ELAN network architecture, s ELAN -A increases the original from 4 times channel to 8 times; that is, the high channel has stronger feature expression ability. However, E-ELAN does not use the method of summing residuals but adopts a stacking method. There is no doubt that the number of calculations is larger, but the representational power is stronger. The head structure is on the 75th, 88th, and 101st layers. The output feature maps have 3 different sizes, and the number of image channels are adjusted for the output features of the different scales through the reparameterized structure REP layer, and they are converted into bounding boxes, categories, and confidence information. Then, the convolutional layer is used as the detection head to perform downsampling to realize the multiscale detection of large, medium, and small targets. Figure 2 shows the structure diagram of each module. 

### 2.2. Improved YOLOv7 Network Model

The improved YOLOv7 model’s architecture is depicted in Figure 3. It involves incorporating the SPPCSPC_Group module to replace the original SPPCSPC module and reducing the parameters and computations. Additionally, the K-means clustering algorithm is utilized for anchor regeneration, leading to faster model convergence. The Ghost Conv module serves as an improved feature extraction network, enhancing the detection while minimizing the parameters and computations. Finally, the CBAM convolution module is introduced to replace the last layer of the backbone network, improving the network’s learning of semantic information and detection accuracy.

### 2.3. SPPCSPC_Group: Group Space Pyramid Pooling Module

YOLOv7 adopts the idea of SPP. First, the feature map is convolved three times, and then the pooling of 5 × 5, 9 × 9, and 13 × 13 is performed, and the feature map of 5 × 5 and 9 × 9 maximum pooling is performed. The ADD operation is spliced with 13 × 13 and the original feature map. After pooling with different kenel_sizes, the feature fusion of different receptive fields is realized, and then after 2 convolutions, it is spliced with the feature map that was convolved once. Figure 4 shows a diagram comparing the SPPCSPC and SPPCSPC_Group structures. The role of the SPPCSPPC is to effectively avoid the image distortion caused by cropping and zooming operations on the image area, and it further solve the problem of the convolutional neural network for image-related repetitive feature extraction, greatly improving the speed of generating candidate frames and saving calculations. Cost, also leads to an excessive amount of parameters in the original SPPCSPC module. To address the above challenges, this paper introduces the SPPCSPC_Group module, which replaces all seven convolution modules in the original SPPCSPC with grouped convolutions. Each convolution module is divided into four groups, offering several advantages. Firstly, efficient training is achieved, as convolutions are processed in parallel with different GPUs. Secondly, grouped convolutions reduce model parameters, as the number of filter groups increases. Finally, they may provide superior models compared to standard full 2D convolutions. The improved SPPCSPC_Group module ensures detection effectiveness while reducing parameters and computations. Figure 5 is a comparison between the original CBS convolution module and the improved CBS-G convolution module.

### 2.4. K-Means

The implementation of the YOLO series of algorithms needs to traverse the preset pixel frames in the image, retain the best pixel frames, and fine-tune them. The above preset pixel frames are called anchor frames. By default, YOLOv7 uses the K-means algorithm to cluster the anchor boxes generated with the COCO data set, and it uses the genetic algorithm to adjust the anchor boxes during the training process, which is the first thing that needs to be done before the class is to initialize the k cluster centers. This makes the selection of the cluster centers in the K-means algorithm highly random and local.

The YOLO series of algorithms require the Initialization of cluster centers (i.e., anchors) before processing the preset pixel frames in the image. This paper improves the K-means clustering algorithm for anchor generation. Instead of using traditional Euclidean distance measurement, 1-IoU (intersection over union) is employed for distance calculation, resulting in improved detection accuracy and effectiveness.

### 2.5. Ghost Conv Module

The current mainstream detection network usually uses ordinary convolution (Conv) for feature extraction, and the structure is shown in Figure 6a. This type of network model uses a large number of convolutional layer stacks, and it further performs convolution operations on each channel of the input layer, which will lead to an increase in the amount of model parameters and calculations. The expressions of ordinary convolution parameter (*P_conv_*) and calculation quantity (*F_conv_*) are:(1)$$P_{conv}=C\times D_{k}\times D_{k}\times N$$
(2)$$F_{conv}=C\times D_{k}\times D_{k}\times N\times D_{f}\times D_{f}$$

In the formula, *C* is the number of input channels, *N* is the output channel, $D_{f}\times D_{f}$ represents the width and height of the output layer, and $D_{f}\times D_{f}$ represents the size of the convolution kernel.

The Ghost module (phantom module, GM), as shown in Figure 6b, is composed of point-by-point convolution and layer-by-layer convolution. The GM first performs point-by-point convolution to obtain the feature enrichment layer map1 and adjusts the number of channels. Each channel of the map1 layer performs layer-by-layer convolution. Finally, it is stacked with the residual edge map1. The number of channels of map1 is the middle channel M, which is equal to one-half of the final output channel number N. It can be seen that the GM uses the residual module to double the stack. Because the stacking of residual modules does not add additional parameters and calculations to the model, the GM can obtain a corresponding number of feature layers while further reducing the resources required for calculations. The expressions of the GM parameter (*P_gm_*) and calculation quantity (*S_gm_*) are as follows:(3)$$P_{gm}=M\times D_{k}\times D_{k}\times C$$
(4)$$F_{gm}=M\times C\times D_{f}\times D_{f}+M\times D_{f}\times D_{f}\times D_{k}\times D_{k}$$

### 2.6. CBAM Attention Mechanism

Since the data set in this paper is a cement pavement defect data set, such as cracks, potholes, and dividing lines, it usually has the characteristics of many occupied pixels and strong correlation among pixels. 

The backbone architecture of YOLOv7 is a deep convolutional network. It captures feature information by setting up multiple 3 × 3 convolutional layers. It has shortcomings such as locality and translation invariance. When establishing model feature relationships, such as long-distance pixel relationships, they are easy to lose as the model depth increases. Therefore, in order to improve the recognition ability of target features, an attention mechanism was added to the last feature layer output with the backbone network. The specific structure is shown in Figure 7. After the layer passes through CBAM, a new feature map will be generated, and the attention weight in the channel and spatial dimensions will be obtained, which will deepen the relationship between the feature in the channel and space, as well as play a key role in extracting the effective features of the target.

The CBAM attention mechanism is a lightweight convolutional attention module that combines the channel and spatial attention mechanism modules. As shown in Figure 7, CBAM consists of two modules, namely, the channel attention module and spatial attention module. The force module allows the network to focus on locations rich in contextual information in the entire image.

#### 2.6.1. Channel Attention Module

The input feature maps are subjected to global maximum pooling and global average pooling based on width and height to obtain two 1 × 1 × C feature maps and then sent to a two-layer neural network. The number of neurons in the layer is C/r (r is the reduction rate), the activation function is Relu, and the number of neurons in the second layer is C. This two-layer neural network is shared. Then, the output features are summed based on element-wise and activated with sigmoid to generate the final spatial attention feature. Finally, the spatial attention feature and the input feature map are multiplied element-wise to generate the input features required by the channel attention module.

#### 2.6.2. Spatial Attention Module

The feature map output by the channel attention module is used as the input feature map of this module. First, a channel-based global maximum pooling and global average pooling are performed to obtain two H × W × 1 feature maps, and then these two feature maps are concatenated based on the channels. After a 7 × 7 convolution operation, the dimensionality is reduced to 1 channel, that is, H × W × 1. Then, spatial attention features are generated through sigmoid. Finally, the feature is multiplied using the input feature of the module to obtain the final generated feature.

## 3. Experiment Materials

The overall flow chart of the experiment is shown in Figure 8.

### 3.1. Experiment Environment

This experiment adopted the PyTorch training framework, took Win10 as the system environment, the GPU model was NVIDIA V100, the memory size was 32 GB, the deep learning environment was Python 3.8, the framework was Pytorch 1.10.0, and the GPU acceleration was CUDA10.2. Because hyperparameters have a great impact on the performance of deep learning models, here we conducted a set of experiments on the batch size, as shown in Table 1. The experimental results show that the model worked best when the batch size was set to 16, and the mAP reached 90.7, so we set the batch size to 16. Table 2 shows the parameter settings of the network model. Table 1 shows the influence of Batch_size on the training process of the YOLOv7 model.

### 3.2. Evaluation Index

In order to evaluate the quality of the model for road surface defect recognition and detection results, the evaluation criteria were precision, recall, mean average precision (*mAP*), and frames per second (*FPS*), as shown in Formulas (5)–(9).
(5)$$P=\frac{TP}{TP+FP}$$
(6)$$R=\frac{TP}{TP+FN}$$
(7)$$mAP=\frac{1}{m}\sum AP\left(i\right)$$
(8)$$FPS=\frac{n}{T}$$
(9)$$AP=\int_{0}^{1}PR\;dr$$

In Formulas (5)–(9), *TP* is the number of correctly recognized detection frames, and *FP* is the number of incorrectly recognized detection frames. *FN* is the number of correct targets not detected; m is the number of detected categories; *AP* is the area under the *PR* curve, reflecting the quality of the model’s recognition of a certain category; *n* is the number of pictures processed by the model; and *T* is the consumption time.

### 3.3. Data Collection and Processing

This experiment used the cement concrete pavement defect image data set collected by the author. The method of collection was to fix a mobile phone at the position of the sun visor of a car’s co-pilot and to take pictures of various defects on the road as the car moved forward. Finally, 2290 images were obtained after screening. Using an open source image labeling tool, LabelImg, 2290 photos were labeled. The images of defects were divided into a training set and a verification set at a ratio of 8:2 of which 1780 were in the training set, and 510 were in the verification set. Table 3 is a statistical table of various quantities. Figure 9 shows an example of a sample image.

## 4. Results and Discussion

### 4.1. Introduction of SPPCSPC_Group Group Space Pyramid Pooling Module

From the experimental results presented in Table 4, it is evident that the introduction of the SPPCSPC_Group module significantly improves the accuracy, reaching 91.2%, while reducing the computations and parameters by 4.3% and 15.4%, respectively. The model demonstrates enhanced efficiency by reducing the parameter count and computational complexity while ensuring accurate detection.

### 4.2. Reset a Priori Frame Comparison Experiment

The anchor processing mechanism of YOLOv7 recalculates the anchor when the anchor in the configuration file calculates the most likely recall rate (best possible recall, BPR), which is less than 0.98. The maximum value of BPR is 1. If the BPR is less than 0.98, the program will automatically learn the size of the anchor according to the label of the data set. The calculated initial anchor frame size is shown in Figure 10, and the comparison shows that the initial anchor frame size distribution in (b) was more uniform and representative.

As can be seen from Table 5, it was found through the experiments that the ratio of the anchor output to the matched target frame during training using the Euclidean distance K-means algorithm was 3.67, the BPR was 0.9932, and the K-means clustering output was represented by (1-iou) as distance The ratio of the anchor to the matched target frame reached 4.2, the BPR reached 1, the detection accuracy was slightly improved, reaching 91.8%, and the reasoning speed also improved; so the experiment shows that the K-means (1-iou) clustering effect is better than the original K-means algorithm.

### 4.3. Effect of Adding GhostConv on the Network with Different Convolutional Layers

As can be seen from Table 6, through experiments it was found that adding the GhostConv parameters and calculations in the 26th to 49th layers of the backbone architecture was much lower than adding it in other positions, and the accuracy was almost the same. Considering that it was used in the 26th to 49th layers of the model of the GhostConv module, the effect of the whole model was the best.

### 4.4. Adding the CBAM Attention Mechanism

In order to improve the recognition ability of target features, an attention mechanism was added in the last feature layer output with the backbone network. After the layer passes through the CBAM, a new feature map will be generated, and the attention weight in the channel and space dimensions will be obtained, and the connection among features in the channel and space will be deepened. It can be seen from Table 7 that although the addition of the CBAM attention mechanism slightly increased the number of calculations and parameters, the accuracy improved, and the largest effect of adding the CBAM attention mechanism was to improve the network’s learning of global information and increasing the obtainment of information. Rich semantic information plays a key role in extracting effective features of the target.

### 4.5. Ablation Experiments

In order to verify whether the improvements proposed in this paper are effective, a set of ablation experiments was designed for comparative analysis. To ensure the accuracy of the experiments, the same parameters were used in the training process, and the experiments were carried out on the self-built data set. This part of the experiment explored the impact of the ten improved methods on the network model. Among them, “×” represents the improvement points that are not included in the list, and “√” represents the improvement points that are included in the list. The plotted data are shown in Table 8. We conducted ten sets of experiments, added different modules, and used the average accuracy map, number of calculations (Flops/G), number of parameters (Params/M), model size (Size/MB), and inference time (FPS) as indicators to be measured and compared.

Table 8 shows the results of the ablation experiments. The combination of the Ghost Conv convolution module and the improved K-means algorithm did not enhance the experimental accuracy significantly, but it resulted in significant reductions in the calculations and parameters. The YOLOv7+GC, YOLOv7+GS, YOLOv7+GKC, and YOLOv7+GKS experiments all experienced slight accuracy reductions; however, they achieved substantial reductions in the calculations and parameters. The YOLOv7+GKCS experiment demonstrated the best combination, with an improved detection accuracy while also reducing the number of parameters and calculations by 29.3% and 14.4%, respectively. These results were attributed to the GhostConv and SPPCSPC_Group modules, which brought accelerated reasoning and lightweight deployment capabilities to the model. Additionally, the improved K-means clustering algorithm and CBAM module significantly contributed to the model’s enhanced detection accuracy.

In order to further observe the effect of YOLOv7 and YOLOv7+GKCS on the verification set, we show their P-R curves, as shown in Figure 11 below.

Judging from the average accuracy of the verification set, the algorithm proposed in this paper is better than the original algorithm on the verification set, which just fulfills the research purpose of this paper to compress the network model while ensuring the detection accuracy remains unchanged.

In order to test the actual effect of the above models, the four models with the best detection effects in the ablation experiment were selected for detection with the following images. These images were all from the training set and the verification set to ensure the reliability and effectiveness of image inspection. Figure 12 shows the detection effect of the above four groups of models on various defects.

As shown in Figure 12, the recognition accuracy of the algorithm in this paper reached 96%, 93%, 90%, and 97% in the recognition of cracks, broken plates, repairs, and boundaries, which is 1%, 9%, 8%, and 1% higher than the original model, and the detection speed reaches 80 FPS. The average accuracy of the improved model did not improve much, but the actual detection effect was better than the original model, and the best accuracy increased by 9% compared with the original model. It can be seen that the algorithm in this paper has a good recognition effect on multitarget and multicategory road surface defects and can effectively detect road surface defects with hidden features. 

### 4.6. Comparison Experiments of Different Network Models

In order to verify whether the algorithm in this paper has advantages using other data sets, experiments are conducted on the public data sets RDD2022-Japan and RDD2022-US. To ensure the reliability of the experiments, they were all tested on the same device and under the same parameters. Table 9 is the comparison of the experimental results of the different data sets.

It can be seen from the experimental results that the average accuracy of the YOLOv7+GKCS algorithm on the RDD2022-Japan data set reached 70.5%, which is 3.3% higher than that of the original YOLOv7. On the RDD2022-US data set, the average accuracy of the YOLOv7+GKCS algorithm in this paper reached 66.4%. This is 2.1% higher than the original YOLOv7. It can be seen that the method we proposed is not lacking in performance when it is extended to different image conditions.

### 4.7. Comparison of the Experiments of the Different Network Models

In order to further verify the effectiveness of the algorithm proposed in this paper, using the same training configuration and data set, other network models and the improved YOLOv7 network model in this paper trained the data set under the same parameters. It can be seen from Table 10 that the algorithm this paper had great advantages compared with mainstream target detection algorithms. This is because we replaced the original SPPCSPC with SPPCSPC_Group, which can reduce the number of parameters and calculations while ensuring the detection effect; second, we added the Ghost Conv module to greatly compress the network model without increasing the detection accuracy so as to achieve a lightweight model; then, we reset the clustering prior frame and added the CBAM attention mechanism to make up for the detection accuracy that was reduced when adding the Ghost Conv module; so the improved YOLOv7 algorithm has further improved detection accuracy, and the model size and number of parameters are also reduced. It has been greatly optimized and also meets the requirements of real-time detection. The requirements provide the possibility for the model to be deployed on edge terminal devices. Therefore, the algorithm in this paper is more efficient than other algorithms, and it is suitable for road defect detection.

## 5. Conclusions

In order to solve the problem that existing pavement defect detection models cannot balance detection accuracy and detection speed, the number of parameters is large, and it is difficult to deploy edge terminal devices with limited computing resources. In order to solve these problems, this paper proposes a pavement model based on the improved YOLOv7 defect detection lightweight model. It mainly includes the following four improvements: first, the group space pyramid pooling module SPPCSPC_Group was introduced to replace the original SPPCSPC module to reduce the number of parameters and calculations; second, the K-means clustering algorithm was used to recreate the anchor generation; third, the Ghost Conv module was introduced; fourth, the CBAM convolution module was introduced to replace the last layer of the backbone network. The experimental results show that compared with the original model, the average accuracy rate of the improved model reached 91%, and the accuracy rates of the broken plate and repaired models increased by 9% and 8%, respectively, compared with the original model. The number of calculations and parameters decreased by 14.4% and 29.3%, respectively, the model size reduced by 29.1%, and the FPS was as high as 80. The improved YOLOv7 reduces the parameters and computations of the model while maintaining high accuracy so as to enable our method to be deployed on edge terminal devices with limited computing resources.

This paper presents an algorithm that effectively reduces the model’s parameters and computations while maintaining high detection accuracy. The proposed lightweight model can efficiently and accurately recognize asphalt pavement defect images, making it suitable for deployment on edge terminal equipment. Nonetheless, real-world road scenes pose complexities, such as defects being ignored due to insufficient lighting, occlusions, and challenging weather conditions. Future research will focus on augmenting the data set to include occluded and weather-specific defect photos, enhancing the model’s ability to detect targets in real scenarios, increasing its practical applicability.

## Figures and Tables

**Figure 1 sensors-23-07112-f001:**
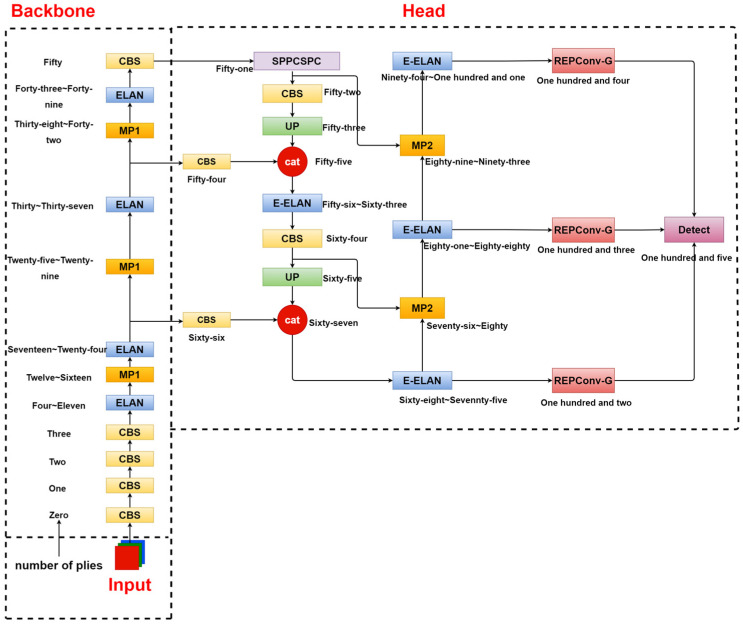
YOLOv7 network structure.

**Figure 2 sensors-23-07112-f002:**
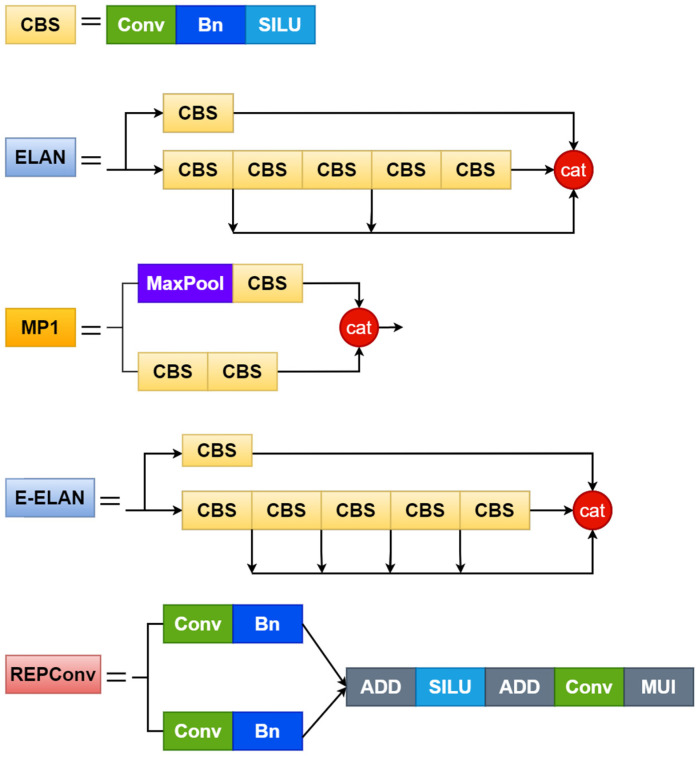
Network structure of each module.

**Figure 3 sensors-23-07112-f003:**
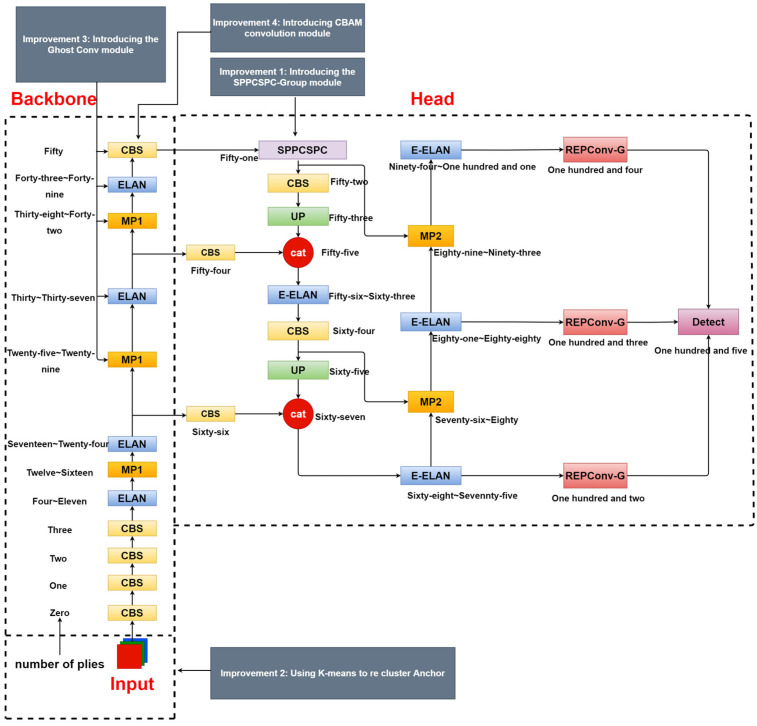
Improved YOLOv7 Network Model.

**Figure 4 sensors-23-07112-f004:**
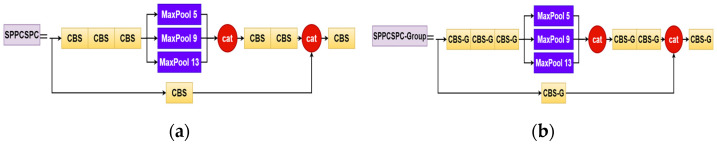
(**a**) SPPCSPC structure diagram; (**b**) SPPCSPC_Group structure diagram.

**Figure 5 sensors-23-07112-f005:**
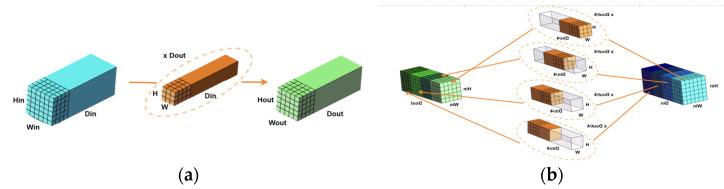
(**a**) CBS convolution module; (**b**) CBS-G convolution module.

**Figure 6 sensors-23-07112-f006:**
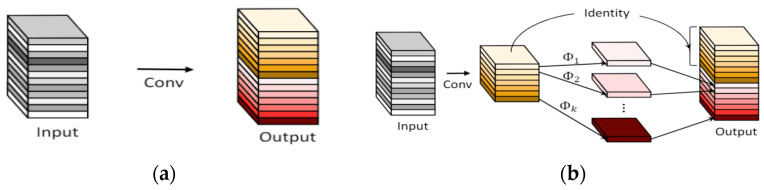
(**a**) Convolutional module; (**b**) Ghost module.

**Figure 7 sensors-23-07112-f007:**
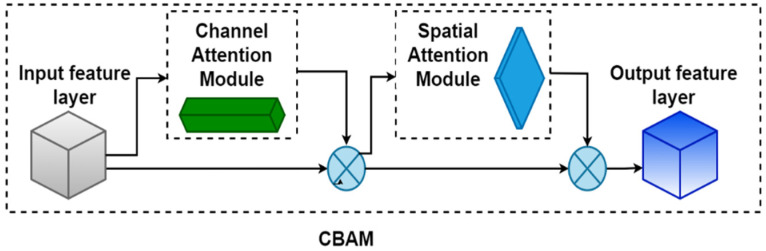
CBAM module.

**Figure 8 sensors-23-07112-f008:**
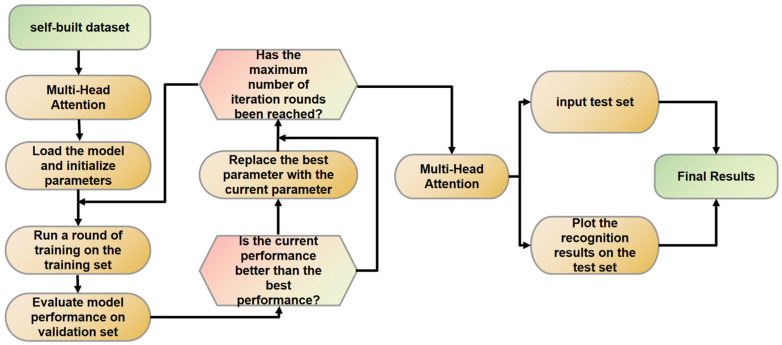
Overall flow chart of the experiment.

**Figure 9 sensors-23-07112-f009:**
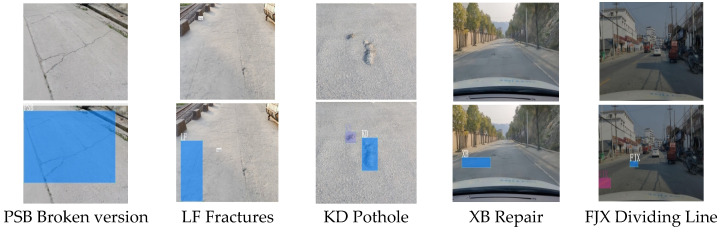
Sample image examples.

**Figure 10 sensors-23-07112-f010:**
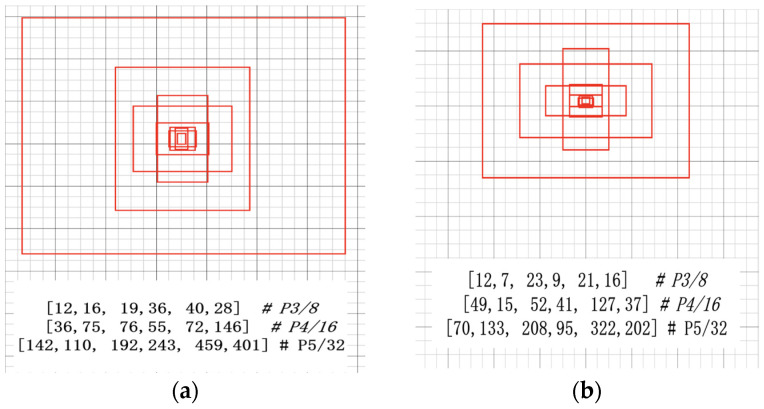
(**a**) Original K-means anchor frame; (**b**) K-means anchor frame.

**Figure 11 sensors-23-07112-f011:**
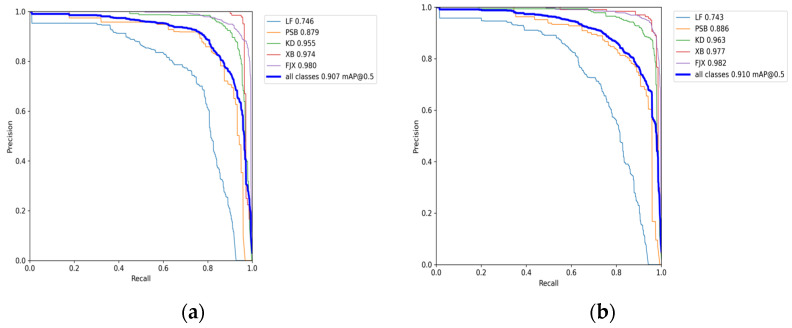
(**a**) P-R curve graph of YOLOv7 on the verification set; (**b**) P-R curve graph of YOLOv7+GKCS on the verification set.

**Figure 12 sensors-23-07112-f012:**
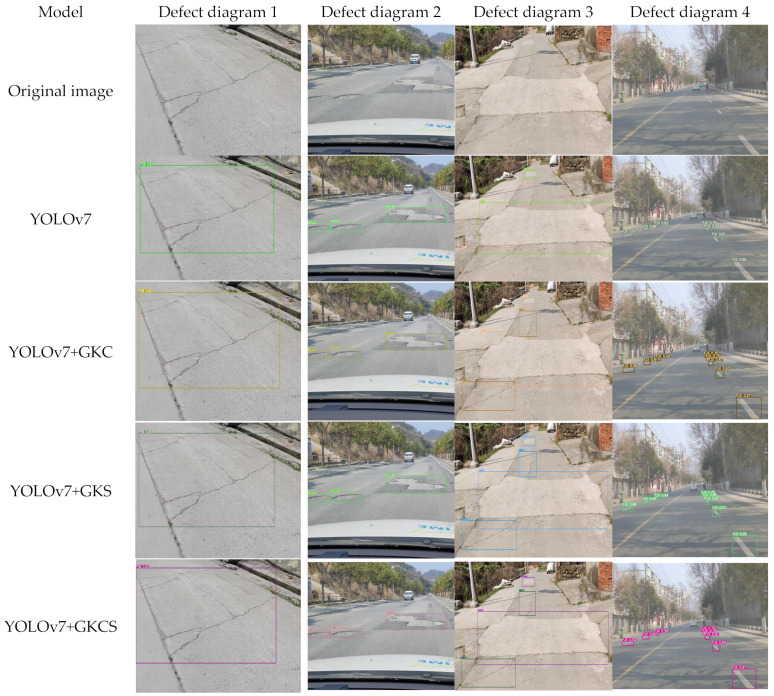
Comparison of the detection effects of the four groups of models.

**Table 1 sensors-23-07112-t001:** The influence of hyperparameters on the training process of the YOLOv7 model.

Batch_Size	mAP%	FPS	Flops/G	Params/M
8	90.6	79	104.8	37.21
16	90.7	80	104.8	37.21
32	90.4	80	104.8	37.21

**Table 2 sensors-23-07112-t002:** Network model parameter settings.

Parameter Settings	Details
Batch_size = 16	Number of batch processes is 16
Epoch = 150	Training 150 rounds of data
Learing_rate = 0.01	Initial learning rate of 0.01
SGD	Optimizer
Weight_decay = 0.0005	The weight decay coefficient is 0.0005

**Table 3 sensors-23-07112-t003:** Statistical table of the number of various types of labels.

Tags	Category	Number
PSB	Broken version	623
LF	Fractures	1423
KD	Pothole	1185
XB	Repair	605
FJX	Dividing Line	2626

**Table 4 sensors-23-07112-t004:** Replacement of SPCSPC_Group comparison experiment.

Model	Layer	mAP%	FPS	Flops/G	Params/M
SPPCSPC	Layer = 51	90.7	77	104.8	37.21
SPPCSPC_Group	Layer = 51	91.2	78	100.3	31.51

**Table 5 sensors-23-07112-t005:** Effect of resetting the a priori frame on the network’s performance.

Algorithm	Dimension Value	mAP%	FPS	Ratio of Prior Box to Target Box	BPR
K-means	[12,16, 19,36, 40,28] [36,37, 76,55, 72,146] [142,110, 192,243, 459,401]	90.7	77	3.67	0.9932
K-means (1-iou)	[12,7, 23,9, 21,16] [49,15, 52,41, 127,37] [70,133, 208,95, 322,202]	91.8	79	4.2	1

**Table 6 sensors-23-07112-t006:** Effect of adding the GhostConv on the network with different convolutional layers.

Attentional Mechanisms	Layer	mAP%	FPS	Flops/G	Params/M
Conv	Layer = 50	90.7	77	104.8	37.21
CBAM	Layer = 50	90.9	78	105	37.34

**Table 7 sensors-23-07112-t007:** Impact of adding CBAM attention mechanism on the network’s performance.

Attentional Mechanisms	Layer	mAP%	FPS	Flops/G	Params/M
Conv	Layer = 50	90.7	77	104.8	37.21
CBAM	Layer = 50	90.9	78	105	37.34

**Table 8 sensors-23-07112-t008:** Ablation experiments.

Model	Ghost Conv	K-Means	CBAM	SPPCSPC_Group	mAP%	FPS	Flops/G	Params/M	Size/MB
YOLOv7	×	×	×	×	90.7	77	104.8	37.22	71.369
YOLOv7+GK	√	√	×	×	90.7	68	93.6	31.37	60.281
YOLOv7+GC	√	×	√	×	90.4	64	94.2	32.02	61.508
YOLOv7+GS	√	×	×	√	90	72	89.5	26.19	50.379
YOLOv7+GKC	√	√	√	×	90.5	72	94.2	32.02	61.508
YOLOv7+GKS	√	√	×	√	89.4	72	89.1	25.67	49.406
YOLOv7+KC	×	√	√	×	90.8	73	105	37.35	71.623
YOLOv7+KS	×	√	×	√	91.7	79	100.3	31.52	60.494
YOLOv7+CS	×	×	√	√	90.9	78	100.5	31.65	60.748
YOLOv7+GKCS	√	√	√	√	91	80	89.7	26.32	50.633

**Table 9 sensors-23-07112-t009:** Comparison of the experimental results of the different data sets.

Data Set	Model	mAP%	FPS	Flops/G	Params/M
RDD2022 Japan	YOLOv7	67.2	74	104.8	37.21
RDD2022 Japan	YOLOv7+GKCS	70.5 (+3.3)	78	89.7	26.32
RDD2022 US	YOLOv7	64.3	65	104.8	37.21
RDD2022 US	YOLOv7+GKCS	66.4 (+2.1)	74	89.7	26.32

**Table 10 sensors-23-07112-t010:** Comparison of the experiments of different network models.

Model	Backbone Network	mAP%	FPS
Faster R-CNN	Resnext101	89	40
YOLOv3	CSPdarknet	78	60
YOLOv4	CSPdarknet	85	70
YOLOv5	CSPdarknet	90	72
YOLOv7		90.7	77
YOLOv8	CSPdarknet	88	78
Ours		91	80

## Data Availability

The data in this study are available upon request from the corresponding author.

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
