# Peer review of "Lightweight Model for Pavement Defect Detection Based on Improved YOLOv7"

_sensors, 2023, doi:10.3390/s23167112_

Round 1

Reviewer 1 Report

In this study, the author proposes a road defect detection model based on improved YOLOv7. The reviewer believes that the manuscript still has the following problems.

1、At present, there are more advanced methods for target detection and Segmentation. For example: (1) Automatic Detection and Counting System for Pavement Cracks Based on PCGAN and YOLO-MF. DOI: 10.1109/TITS.2022.3161960.  The reviewer recommends that the authors analyze the methods of the above literature in manuscript.

2、In the manuscript, many images are not clear (e.g. Figure 1, Figure 3, Figure 5), please improve them.

3、The hyperparameters has great influence on the performance of deep learning model. The reviewer suggests discussing the choice of hyperparameters.

4、In Section 3.2, formula (7) gives the definition of mAP. However, the calculation formula of AP is lacking. Please give the AP calculation formula.

5、Table 2 shows the composition of the data set. As can be seen from Table 2, the amount of data for various types of defects is unbalanced. Please analyze the impact on the model performance.

6、Figure 11 is unsightly. In addition, the font in Table 7 is not Times New Roman.

7、In Section 4.6, the reviewers suggest adding comparative experiments with other advanced methods. In addition, in section 4.6, the reviewers suggest an analysis of the reasons for the excellent results of the proposed method.

Moderate editing of English language required

Reviewer 2 Report

Issue:

The author proposes an improved YOLOv7 algorithm to address the low detection speed and accuracy issues in previous road surface defect detection methods.

Innovation:

In the improved YOLOv7, the author replaces the original network structures with four modules: SPPCSPC_Group, K-means (1-iou), Ghost Conv, and CBAM. These modules reduce the network parameters, improve model inference speed, and enhance detection performance.

Improvements:

The author made improvements to the existing detection algorithms within the context of road surface defect detection. However, these improvements are not described in relation to the engineering background, and the level of innovation in the improvements is insufficient.

Experiments and Results:

In the experimental section, the author selects five common road surface defect datasets for training and detection. The author also conducts ablation experiments, comparing and analyzing the results of each improvement in terms of mAP, FPS, and parameter quantity. However, the detection results for each type of defect are not provided.

Text and Figures:

The structure diagrams of the YOLOv7 model and the modified model in the paper need to be redrawn, and the font size for descriptive text is too small.

Conclusion:

The author improves the YOLOv7 detection algorithm within the context of road surface defect detection, resulting in improved accuracy and speed. However, from an engineering perspective, the author's improvements to the algorithm are not sufficiently integrated with specific engineering issues. From an algorithmic perspective, the improvements made by the author are not innovative enough.

The English writing of this article is acceptable, but some formatting correction is still needed, especially for the Figures.

Reviewer 3 Report

This paper introduces a so-called group space pyramid pooling module SPPCSPC-Group to replace the original SPPCSPC, the improved YOLOv7 has reduced the number of parameters and calculations of the model while maintaining high accuracy, compared with other algorithms. Focusing on the core idea of SPPCSPC-Group, here only the advantages are provided by the authors, but the rationale of the algorithm was not mentioned at all. However, the group’s idea has already been reported in several works and open-access websites. The idea of introducing K-means is poor. The convolution block attention module (CBAM) has already been reported for several years. Moreover, many grammar errors and Chinglish problems are not addressed in this work.

Many grammar errors and Chinglish problems are not addressed in this work.

Reviewer 4 Report

This paper proposes a pavement defect detection light weight model based on improved YOLOv7.

The article is interesting and publishable, but changes are needed:

1. There are shortcuts in the introduction, e.g. SPPCSPC, CBAM. Can these shortcuts be replaced with names? Shortcuts should be used after explaining their meaning.

2. Figures 1, 3, 5a are of poor quality and the descriptions in the figure cannot be read. Please post a better quality figures.

3. Is it possible to include a diagram showing the course of research in the method?

4. The conclusion lacks descriptions of the numerical results. Please provide such information.

5. In the conclusion, they should be related to previous studies of this type. What is new about this article compared to other works?

After making these changes, the article is ready for publication.

Reviewer 5 Report

The authors propose a “Lightweight model for pavement defect detection based on improved YOLOv7”, which attempts to address the pavement defect detection using devices with limited computing resources.

The authors correctly introduce the problem and lacks good literature review of today’s algorithms applied to the topic. The authors could also compare with other deep learning algorithms, like vgg-x, res-net, that are used for image classification.

After reading the paper I would like the authors comments the following notes:

-The authors compare the standard Yolo v7 with the introduced modifications specifically to reduce parameters and other optimizations. According to table 7 there is no significative reduce in computer demands (FPS) and the mAP is practically the same. Why don’t use the Yolo on a vision cloud server processing?

-Why did the authors not use a public well characterized and balanced dataset? After train, test, and validation, then in the second experimental the author could use their concrete images.

-The dataset used are not well characterized and with the correct balance between different five classes, according to table 2.

-The present research work is interesting in the computer vision field, but from the paper itself it’s not an innovative evolution, and to conclude what authors conclude.

As finally, can your proposed method be generalized to different image conditions with no lack of performance?

As a conclusion of the review, the authors are invited to address my questions preferably. The paper is well written and for this reason my recommendation is that it may be accepted.

Round 2

Reviewer 1 Report

All comments were addressed.

Author Response

Thank you very much for your positive and constructive comments and suggestions on our manuscript, and we very much hope that the revised manuscript will be accepted by you. Thank you and best regards.

Reviewer 2 Report

The author has implemented revisions in response to their feedback, and I currently possess no further inquiries.

Some minor grammar and formatting revisions are still needed.

Author Response

Thank you again for your positive and constructive comments and suggestions on our manuscript. We checked the manuscript many times and asked an editor from the United States to check and help us to correct the grammar and formatting problems in the article. For your convenience, the revised sentences are marked in green. We very much hope that the revised manuscript can be accepted by you. Thank you and best regards.